# Consumer Preferences for Origin and Organic Attributes of Extra Virgin Olive Oil: A Choice Experiment in the Italian Market

**DOI:** 10.3390/foods10050994

**Published:** 2021-05-02

**Authors:** Matteo Carzedda, Gianluigi Gallenti, Stefania Troiano, Marta Cosmina, Francesco Marangon, Patrizia de Luca, Giovanna Pegan, Federico Nassivera

**Affiliations:** 1Department of Economics, Business, Mathematics and Statistics (DEAMS), University of Trieste, 34100 Trieste, Italy; matteo.carzedda@units.it (M.C.); marta.cosmina@deams.units.it (M.C.); patrizia.deluca@deams.units.it (P.d.L.); giovanna.pegan@deams.units.it (G.P.); 2Department of Economics and Statistics (DISES), University of Udine, 33100 Udine, Italy; stefania.troiano@uniud.it (S.T.); francesco.marangon@uniud.it (F.M.); 3Department of Agricultural, Food, Environmental and Animal Sciences (DI4A), University of Udine, 33100 Udine, Italy; federico.nassivera@uniud.it

**Keywords:** choice experiment (CE), extra virgin olive oil (EVOO), willingness to pay (WTP), country of origin, organic food, consumer preferences, sustainable food system

## Abstract

The paper investigates Italian consumers’ behavior towards characteristics of extra virgin olive oil, in particular organic production methods and geographical origin. On the basis of the existing literature, the concepts of sustainability of food systems, diets, and the olive oil supply chain are analyzed. A choice experiment (CE), using a face-to-face questionnaire with over 1000 participants, was conducted to quantify the willingness to pay (WTP) for these two attributes. Findings show positive preference for origin attributes, while the organic attribute is not highly valued. The article also offers some perspectives on future research to improve the competitiveness and sustainability of the Italian olive oil supply chain.

## 1. Introduction

The olive oil system plays a central role in the sustainability of the food system that underlies the Mediterranean diet patterns, for both environmental and socioeconomic aspects [1].

Therefore, it is particularly interesting to analyze consumer behavior for some sustainability attributes of extra virgin olive oil (EVOO), the highest-quality product in the olive oil supply chain.

In fact, such consumers’ preferences are one of the main drivers for the transition from current eating patterns to a more sustainable one. Nevertheless, consumers’ preferences concern not only the agricultural and production processes, but a complex set of product attributes [2]. Among these are the perceived sensorial quality of olive oil, such as taste, acidity, and fragrance, but also color, country of origin, geographical indication (GI), and the use of olives from organic farming [3].

Unlike other types of olive oil, such as ordinary, refined, lampante, and olive-pomace oils, virgin oil is exclusively obtained through mechanical extraction processes, namely, washing, decantation, centrifugation, and filtration [4]. Superior-quality virgin oils, in terms of raw materials and quality of the production process, are classified as EVOO and are particularly valued for their organoleptic and nutritional properties [5], such as low acidity levels (below 0.8%, according to European standards) and high content of monounsaturated fat and polyphenolic, antioxidant, and anti-inflammatory compounds [6]. In addition to its intrinsic, sensory, and health attributes, the value of EVOO is further enhanced by its potential to promote multifunctional and sustainable agricultural models, which is particularly true and beneficial for traditional olive-tree-growing regions [7]. In this context, the transition from current eating patterns to more sustainable ones depends on consumers’ preferences for this complex of attributes [2].

This specific role related to sustainability of the olive oil system depends on the general framework in which the idea of sustainability is developed. Sustainability has become more and more used in different strategic documents, policies, and development plans at the international, national, and local levels. Among these are the so-called “Brundtland Report” [8] and “2030 Agenda” of the United Nations (UN) [9], and inside the European Union (EU), the Common Agricultural Policy (CAP), the Green Deal [10], and the “Farm to Fork” European strategy [11].

Within the agrifood sector, the concept of sustainability is historically closely related to organic production, although sustainability can refer to a wider range of agricultural elements and practices. These include precision agriculture, organic farming, agroecology, agroforestry and stricter animal welfare standards, carbon managing and storing, and adoption of circular economic models [10]. Nevertheless, organic certification remains for the consumer the main recognizable distinctive sign of the environmental sustainability of food.

The International Federation of Organic Agriculture Movements (IFOAM) defines and constantly updates the Basic Standards for Organic Production and Processing (IBS), the founding principles, definitions, and requirements on which national organic certification schemes, such as the Soil Association Standards in the UK, the USDA National Organic Program, and the Indian National Programme for Organic Production (NPOP), are based [12]. Among the broad array of national and international regulations governing organic certification schemes, Council Regulation No. 384/2007 [13] sets the legal basis for organic farming within the EU and “defines organic production as an integral system of managing and producing food products, which combines the best practices with regard to the preservation of the environment, the level of biological diversity, the preservation of natural resources, the application of high standards of proper maintenance (welfare) of animals and a method of production that corresponds to certain requirements for products manufactured using substances and processes of natural origin” [14] (p. 4). Given the spatial extent of our study, we will from now on refer to the EU organic rules, whose principles, meaning, and visual identity are generally well recognized and acknowledged by European consumers [15,16].

The overall aims of the EU action concern the transition of the European agrifood sector towards a sustainable production and consumption model, also adopting actions to help consumers choose healthy and sustainable diets [10,11].

This objective is consistent with the results of several studies [17,18,19] that link sustainability with healthy diet through the concept of food system, which is recalled by the “Farm to Fork” European strategy.

Sustainable food systems emphasize the role of dietary styles as core links between foods, human health, and nutrition outcomes [20,21,22]. A sustainable food system should generate positive outcomes related to the three dimensions of sustainability. In other words, it should be economically profitable, provide equitable benefits for society, and have a positive or neutral environmental impact [21].

Within the sustainable food system approach, the Mediterranean diet (MD) plays a primary role [23,24,25]. Several findings reveal that the MD pattern demands less soil, water, and energy compared with other consumption patterns, such as the Western dietary patterns and meat-based diet, characterized by high environmental impact [26,27]. Moreover, while the Mediterranean region has been a major food-producing area with a large agro-biodiversity for millennia, environmental alterations may threaten the local food system capacities to ensure food and nutrition security [28]. In fact, the Mediterranean region is facing massive environmental changes: land use and degradation, water scarcity, environment pollution, biodiversity loss, and climate change [28,29].

Therefore, the notion of MD has undergone a progressive evolution over the past decade: from a healthy dietary pattern to a sustainable diet model and to a catalyst for a resilient strategy of the Mediterranean area [30,31,32,33].

However, the transition from less sustainable, currently widespread diet in most European countries, also in the Mediterranean area, towards a more sustainable MD requires substantial changes in consumers’ values, education, and choices. According to the definition of food system, consumer behavior, together with food supply and food environments, is an important driver that determines the nutrition and connection to health [2,34].

In fact, the perception among consumers of the MD as a healthy diet and the image of olive tree as a symbol of the Mediterranean lifestyle have pushed the demand for typical local foods of the Mediterranean area, EVOO in particular [35]. Olive oil, especially EVOO, has become one of the most important and recognizable symbols of the MD patterns [33], and is conventionally linked to the concept of well-being, not only in Italy, but also worldwide.

Therefore, domestic consumption of olive oil has been continuously growing for a decade and is expected to grow in nonproducing countries as well, while a strong demand in both traditional and new markets will favor an increase of exports from producer countries [36], generating positive economic impacts on the Mediterranean area, and EU countries in particular. Spain, Italy, and Greece alone produce some 70% of the global olive oil supply. In the years 2015–2019, EU production represented 69% of world production, while the provisional figure for 2019/2020 shows a share of 60%, and the forecasts for 2020/2021 indicate an increase of up to 68% of the world production of olive oil. In the same period, the Mediterranean countries of the EU consumed more than half of the world production [36]. The EU olive oil sector is expected to grow in production capacity by 1.1% per year on average, reaching 2.4 million tons in 2030 (compared with 2 million tons in 2019) [36].

Another process of market and product evolution accompanies this trend. The olive oil market has evolved from a traditionally “bulk” market, which conceived olive oil as a mere commodity, similar to other vegetable fats, to a more customized market, in which quality and sustainability claims are multiplying. Therefore, olive oil is increasingly perceived as a food specialty, similar to wine or other high-quality products [37,38,39,40].

The analysis of the literature findings highlights heterogeneity in olive oil consumption habits not only between traditional new consumer countries, but also across Mediterranean countries (see in particular [3,37,38,39,40,41,42,43,44,45,46,47,48,49,50,51]). For instance, Dekhili et al. [40] point out the relevance of oil color in Tunisia and France, while Ribeiro and Santos [41] focus on Portuguese consumers’ preference for low-acidity oil.

Indeed, contemporary olive oil, in particular EVOO, consumers are increasingly mindful and aware of attributes, such as sustainability, supply chain ethics, and the intrinsic quality of the product; moreover, they show a growing interest in organic production processes and geographical origin of the product and raw material [52,53,54,55,56,57,58,59]. These two attributes concern the environmental and socioeconomic sustainability of the production systems and include both the quality characteristics of the products themselves and ethical aspects: environmental protection and local development [60,61,62,63,64,65,66,67]. It should be noted that organic production and geographical indication (GI) are adequately certified by community standards, hence easily recognizable by EU consumers.

In fact, several authors highlight the relevance of the organic attributes of EVOO and their importance to consumers, in particular, Tsakiridou et al. [62], Liberatore et al. [68], Roselli et al. [69], and Perito et al. [70]. Most of the themes also investigate organic and origin attributes together.

Moreover, recent studies on food choice and consumption demonstrate that consumers pay attention to the country of origin, suggesting that a certain product image reflects the image of the region or country of production [3,71].

To a broader extent, consumer preference for GI attributes of EVOO is investigated, in particular by Di Vita et al. [46], Roselli et al. [69], Tempesta et al. [71], Erraach et al. [72], Finardi et al. [73], Ballco et al. [74], Fotopoulos et al. [75], Perito et al. [76], and Menapace et al. [77]. These studies point out that higher-income consumer groups were more aware of geographical origin certification labels. According to Erraach et al. [72], for Spanish oils, attributes of origin, region of production, and quality directly affect their market potential, while Perito et al. [76] find that olive oil production region is an important driver of choice for Italian consumers. In addition to this, Italian consumers in particular show very high knowledge of and demand for extrinsic attributes, such as place of production, designation of origin, organic certification, and type of processing for extra virgin olive oils [71], all attributes that bridge production sustainability and perception of high quality.

Given these considerations, this paper investigates the attitudes of a sample of Italian consumers towards organic and origin attributes of EVOO using a choice experiment (CE). The discussion of the results provides interesting insights on consumer preference for EVOO attributes and paves the way to further advances in scientific research on competitiveness and sustainability of the Italian olive oil supply chain.

Our findings point out a preference heterogeneity in the information perceived by olive oil consumers, identifying a number of unobserved sources of heterogeneity in their decision process. The results also reveal a strong and positive preference for locally produced olive oil rather than an organic product. These consumer attitudes towards extra virgin olive oil are not in contrast with each other but fall within the framework of a sustainable development model that takes into account not only the environmental dimension but also the socioeconomic one, linked to the local development of the Mediterranean area. This model should link local development strategies with healthy diet goals, finding its keystone in the Mediterranean diet patterns.

## 2. Materials and Methods

Several studies, in recent years, have been carried out in the context of new and diversified trends of EVOO consumer demand [44,45,46,53,55,58], and different surveys have shown that consumers’ choices widely differ with respect to sensory preferences, extrinsic quality signals, experience, purchase motives, perception of supply elements, and socioeconomic characteristics [47,59,60,61,62]. Moreover, the literature results indicate that organic certification, origin of both olives and olive oil, and price are the main extrinsic attributes of EVOO guiding the choice process of consumers [68,69,70,76,77].

A broad meta-analysis by Del Giudice et al. [45] on scientific studies on EVOO consumer preferences published between 1994 and 2014 highlights the influence of origin and its various certifications, as well as brand recognition, on consumers’ choice. Although the results emerging from this literature survey are somewhat heterogeneous, it is still possible to identify common trends, namely, the importance of the country of origin of olives; a growing interest in organic certification; and the importance of trust in the brand, whether it be a traditional long-established producer or a trustworthy private label.

Among the methods used to estimate consumers’ preferences for specific attributes of goods, conjoint hedonic methods, classic hedonic testing, and alternative descriptive approaches are widely used in the recent literature on consumer studies [48,63]. The basic idea behind conjoint analysis (CA) and CE is that public and private goods can be described as a bundle of product attributes; each combination of these characteristics results in a different product, and survey respondents are asked to evaluate these changes [78]. The experimental design of CA and CE allows researchers to estimate the independent effect of each product attribute on product evaluations or product choices by respondents.

In particular, CE is based on Lancaster’s [79] characteristics theory of value in combination with the random utility theory [78]. Therefore, statistical analyses of the responses obtained from CE are used to estimate the marginal values of product attributes, which represent the premium price that consumers are willing to pay for the desired characteristics.

In detail, we estimated the WTP for the attribute level by dividing β coefficients by βprice.
WTP = −β/βprice(1)

With reference to consumer demand, it is necessary to note that this approach means the adoption of the so-called new consumer demand theory [79], and consequently, there exists the operational problem of estimating consumers’ WTP for specific product attributes. As is well known, the Lancaster approach is an evolution of the traditional microeconomic theory of demand, in which the utility of goods is derived from their characteristics (and not from the goods per se); therefore, the utility of product alternatives is a latent construct that only exists in the minds of individual consumers. Researchers cannot observe this directly. Nonetheless, indirect measurement techniques can be used to explain a significant part of the latent utility construct. However, the error component determined by additional unobservable attributes, measurement errors, and variation between individual consumers remains unexplained.

First, this study used a multinomial logit model (MNL) in which consumers are assumed to be homogeneous. Moreover, considering that consumers are widely recognized as heterogeneous in their preferences [80], we used a latent class (LC) model that assumes hidden latent classes for consumers and products. Olive oil can be characterized by different attributes, such as price, origin and environmental certification, and private brand.

This approach combines insights from the characteristics theory of value that assumes that individuals do not derive utility from a product per se, but from a product’s characteristics [79], as well as from the random utility theory (RUT) [81]. RUT models consumers’ preferences among mutually exclusive discrete alternatives by drawing a real-valued score on each of them (typically independently) from a parameterized distribution and ranking these alternatives according to score models.

Consumers typically have only basic knowledge of EVOO, and therefore, information plays an important role. Consequently, the label information and certification logo are important means to convey and ensure the existence of the characteristics desired by consumers. The theoretical basis for this aspect is the economics of information [82,83]. In particular, Akerlof [83] was the first to show that asymmetric information, such as uncertainty about the quality of a good, can cause a market to degenerate into an exclusively low-quality product market.

Therefore, this study applies the CE methodology to the Italian EVOO market to estimate not only the ordinal ranking of preferences of consumers, but also their willingness to pay (WTP) for key product characteristics. To do this, we used data obtained from a field experiment through face-to-face interviews with household consumption decision makers conducted using a dedicated questionnaire.

The structure and contents of the questionnaire were discussed, during its preparation, with some university researchers from different disciplines (marketing, raw materials sciences, and agricultural economics) other than the authors, who teach at the University of Trieste and the University of Udine.

In order to confirm the clarity and understandability of the questionnaire and test the statistical possibilities of the gathered data, a preliminary draft was discussed in a focus group consisting of 10 consumers responsible for their own household food shopping.

Comments and observations gathered during this preliminary study allowed us to update and revise the questionnaire, whose average response time was estimated to be about 15–20 min. Prior to the actual data collection phase, the interviewers were trained in survey administration.

The interviews were conducted outside supermarkets or food shops; therefore, the random sample adopted is not representative of the population and can represent one of the limitations of the present study.

The final version of the questionnaire was organized into three sections. The first one included descriptive information on the respondents’ demographic characteristics and professional background. The second section contained questions on the respondents’ consumption habits. The last section presented the participants with a discrete choice of olive oil characteristics.

The scales adopted for the second section of the questionnaire included qualitative values from “never” to “always” to identify purchasing habits, consumption frequencies, and preferred purchasing channels. To verify the consumers’ knowledge of MD, EVOO, organic farming, and designation of origin, multiple-choice questions were used with only one correct answer.

The first stage of developing a CE involved identifying attributes relevant to our research, and then determining the levels of each of these attributes. In our study, the attributes of interest in the hypothetical olive oil bottles were informed by reviews of literature and interviews with different stakeholders, and discussed within a focus group interview.

Price: price is the traditional economic variable that influences consumer demand in a negative way. The different price levels were chosen based on the actual prices of olive oil as assessed during a store check in January 2018 in food stores in northeastern Italy. Subsequently, three price levels were identified for a 1000 mL EVOO bottle based on a sample of bottles with attributes corresponding to those used for the choice experiment. Therefore, we considered three levels: €4.00, €8.00, €12.00 for a bottle of 1 L EVOO.

Country of origin (COO): among the mandatory EVOO attributes [84], the country-of-origin brand is probably the most recognizable. This attribute points out that the COO is a component of an EVOO brand and adds value to an EVOO purchaser. In traditional producing countries, but also in some other new consuming countries (see the literature cited above), this is a particularly relevant attribute of EVOO. In accordance with EU legislation [84], we considered three levels in hierarchical order of value: (a) 100% Italian olive oils, (b) blend of olive oils of EU origin, and (c) blend of olive oils of EU and not-EU origins.

Geographical indication (GI): The geographical indication of EVOO is another distinctive sign used to identify a product as originating from the territory of a particular country, region, or locality where its quality, reputation, or some other characteristic is linked to its geographical origin. In this context, the term is used to refer to the EU legislation. The EU legislation [85] includes two types of certification: protected designations of origin (PDO) and protected geographical indications (PGIs). These labels certificate that the product is linked to a geographical area, where “protected designation of origin” (PDO) has a stronger quality–geography link and higher qualities than “protected geographical indication” (PGI). Such certifications are important drivers of local development and can therefore be considered attributes of the social and economic dimensions of sustainability. Our CE considered three levels in hierarchical order of value: (a) PDO, (b) PGI, and none (EVOO without GI certification).

Organic: The organic characteristic of food is the main environmental attribute of sustainability. This attribute of EVOO is particularly relevant not only in traditional producing countries but also in some new olive oil-consuming countries (see the literature cited above). According to EU legislation [12], we considered the organic label logo to identify organic certification and to inform respondents about the presence of this attribute. The CE used a dichotomous variable (yes/no) corresponding to the presence (or absence) of organic labelling.

Market leader brand: private labels play a relevant role within the EVOO market, in particular in large-scale retail distribution channels [86]. In particular, in the Italian market there exist numerous famous oil producers with their own brands, even though this is not a straightforward guarantee of the Italian origin of raw materials (see COO attribute), and even the ownership of such companies is often no longer Italian. Some private brand names, leaders in the oil market, are proposed in the questionnaire as examples for the interviewees (Bertolli, Carapelli, Dante, Farchioni, Monini, and Sasso). The CE used a dichotomous variable (yes/no) corresponding to the presence (or absence) of a brand of market leader.

Table 1 provides the description of each attribute and related levels.

The discrete CE captured responses regarding the choice of olive oil bottles.

An orthogonal fractional factorial design was then generated using SPSS^®^ software, with 18 alternatives (or profiles) selected. The profiles were randomly combined into six sets of choice, all presented to each respondent. The alternative bottles proposed to the respondents had no difference in any aspect (color of the bottle, year of production, acidity, etc.), with the exception of the five specific attributes described above. Each respondent was asked to compare three EVOO bottle options and choose the favorite one. In order to simulate a realistic choice context, the opt-out (no choice) alternative was included in the choice sets to grant the consumers the freedom of choice they have in real market situations, where they can also decide not to purchase any bottle at all. In addition, the interviewees were asked to consider the choice tasks as separate, individual situations and to answer each of them.

Figure 1 graphically represents an example of a choice set.

A preliminary pilot study using the same characteristics was carried out in 2016 and ended on January 2017. The results of this survey [87] were used to clarify some questions, insert new ones, and remove others, and in general, to refine the overall questionnaire.

Then, a second face-to-face survey was conducted by administering a questionnaire to citizens in the northeastern part of Italy to determine their preferences for olive oil. The survey was carried out in 2018 and ended on January 2020. A total of 1024 consumers were interviewed. Participants were not offered an honorarium in exchange for their response and time.

## 3. Results

### 3.1. The General Profile of the Respondents and Their Consumption Behaviors

With respect to their socioeconomic characteristics, 60.61% of the participants were female; 45.51% were employed. All relevant age classes were represented in our sample, with 29% of the respondents being between 40 and 55 years old. The average level of education of the sample was relatively high, as 45% of the respondents held a high school diploma.

Regarding oil consumption habits, the respondents’ answers show that over 90% of them mainly consume EVOO.

The analysis of purchasing channels highlights the prevalence of large-scale distribution (51.59%), followed by direct purchase from farms/olive oil producers (28.01%) and consortia or cooperatives of olive oil producers (12.05%). Purchase from retailers or specialized shops represents only 8.35% of the total answers.

Furthermore, with respect to product knowledge, over 90% of the interviewees know the characteristics of the Mediterranean diet and that olive oil is included in this diet pattern.

Approximately 73% of the interviewees stated that they know the characteristics of EVOO, but only just over 50% were able to correctly recognize in a multiple-choice question the characteristics that distinguish this product. In line with the results from previous studies on Italian EVOO consumers [88,89], this element highlights a general recognition of the high quality and wholesomeness of the product, but limited specific knowledge, despite the good level of education of the respondents.

Additionally, just over 52% of the respondents said they read the label on the EVOO bottle. However, this data could be linked to consolidated purchasing habits of the same product, to the greater attention paid to the certification logos possibly present on the bottle, or on the wording of extra virgin olive oil.

Coming to organic and geographical certifications, the results of the survey show that the interviewees generally know the characteristics of organic olive oil (69%) and PDO/PGI olive oil (80%). Moreover, 16% of the respondents buy “often” or “always” organic olive oil, and 63% of them purchase such products at least sometimes. Similarly, 31% of the respondents purchase “often” or “always” PDO/GPI olive oil, while 83% of them buy it at least sometimes.

These answers show a good knowledge and consumer attitudes towards organic and geographical indication certifications of EVOO. These results share a number of similarities with those of Polenzani et al. [90], who associate traditional consumption of EVOO and general knowledge of the product and its main attributes.

Finally, the survey results provide insights into the attributes that the consumers surveyed find most important. They were asked to express the importance of a set of attributes with values between 0 (min) and 10 (max), with the possibility to evaluate different attributes with equal scores. The results are as follows: 27% indicate the Italian origin of olives as the priority attribute in EVOO choice, 14% the regional location of the place of production of olive oil, 11% of consumers the local origin of the product, and 11% the PDO/GPI certification. Only around 7% of the respondents indicate organic certification, and 6% of them the price of the bottle. Our results are consistent with those of previous studies on the EVOO market and consumers’ choice drivers [89,91,92]. Overall, almost 80% of the respondents pay attention to the national, regional, or local (including GI) origin of olives and/or olive oil. It is worth noting that the specific location of the oil mill (e.g., in one specific Italian region: Tuscany, Umbria, or Puglia) does not ensure that the raw materials (olives) come from the same territory. This may be a lack of knowledge on the part of consumers or unintentionally unclear information on the characteristics of the production chain.

### 3.2. Choice Experiment: Statistical Analysis

CE data were analyzed using NLogit4^®^ software. Several empirical models were tested. In addition to the multinomial logit model (MNL) for the main effect and interactions, we further analyzed data using a latent class model (LCM). As only part of the variability in the intensity of the assessment can be associated with measurable socioeconomic characteristics, the LCM was used to reveal the component of heterogeneity associated with unobservable characteristics. This model relaxes the assumption of independence of irrelevant alternatives that result from the MNL. According to Boxall and Adamowicz [93], LCM allows for the random distribution of parameters across the population, capturing preference heterogeneity.

The LCM identifies the utility that a respondent belonging to a particular segment derives from choosing a bottle of olive oil with extrinsic attributes in different contexts. LCM determines the probability of a respondent in a segment to choose a particular alternative, and the choice probability is conditional on class probabilities. As stated by Hu et al. [94], instead of relying solely on standard demographic variables, the LCM uses information derived from respondents’ choices to estimate preferences.

Taking into consideration the log-likelihood function (LL), Akaike information criterion (AIC), Bayesian information criterion (BIC), Hannan–Quinn information criterion (HQIC), and pseudo R-squared indicators (Table 2) as suggested in theory, the three-class model was chosen due to its superior performance.

Table 3 presents the results for the two models (MNL and LCM). Each coefficient (β) indicates the direction and relative importance of an attribute on utility derived by the respondents. In the base model (MNL), not all attributes were statistically significant (*p* < 0.05). Briefly, the price was significant at a 90% confidence level, while the organic attribute was not significant, indicating that this attribute was not important in determining olive oil purchase intentions among the respondents. In addition, the leading brand was significant at 94%.

According to the results of the MNL, Italian-origin olive oil had the highest preference among the respondents choosing a bottle of olive oil (β = 1.82, *p* < 0.05). Olive oil of EU origin (β = 0.83, *p* < 0.05) and PGI certification (β = 0.63, *p* < 0.05) also increased the utility perceived by the respondents, though to a lesser extent. In addition, they preferred a leading brand (β = 0.11, *p* < 0.01). Finally, in contrast to earlier findings [46,72], PDO certification decreased the utility perceived by the respondents (β = –0.28, *p* < 0.05).

### 3.3. Choice Experiment: Consumer Class Definition

The LCM showed various sources of preference heterogeneity in the information perceived by the olive oil consumers, as highlighted by the analysis we obtained for each class. The results revealed that 30% of the respondents belonged to class 1, 46% to class 2, and the remaining 24% to class 3.

Class 1

The coefficients for the respondents belonging to class 1 were significant at a 95% confidence level apart from price, which was not significant, while EU origin and organic attribute were significant at a 90% confidence level. The participants of this group showed a strong preference for Italian olive oil, and they seemed to attribute importance to the presence of a leading brand and to PGI certification. In addition, EU origin and organic attributes were appreciated but to a lesser extent. This group disliked a PDO designation and did not consider price to be an important attribute in purchasing olive oil. Given the statistical insignificance of the price coefficient, the WTP estimation does not make sense.

Class 2

Class 2 associates its olive oil choices with Italian and EU origins and with a PGI denomination. At a lower level, the presence of a leading brand increased the respondents’ utility, while they disliked PDO certification. For this group of respondents, it was possible to look at the WTP, as coefficients were all significant at least at a 90% confidence level with the exception of the organic attribute coefficient. Specifically, the respondents declared that they were willing to pay €13.35 per liter and €11.80 per liter for Italian- and EU-origin olive oils, respectively. The estimated WTP for PGI certification and for a leading brand was €6.69 and €2.67, respectively.

Class 3

The coefficients for group 3 were all statistically significant at 95% or 90% confidence levels apart from the organic attribute. The members of this class had a clear preference for Italian-origin olive oil and PGI certification. EU origin also increased their utility, leading brands and PDO designations. In addition, contrary to our expectations, the coefficient of the price variable for this segment was positive, implying that ceteris paribus, the higher the price, the higher will be the probability of choosing a given olive oil. Although it is plausible to think that a purchase decision could be influenced by price as a signal of quality at least to a threshold price level [95,96], this finding cannot be justified for a rational economic agent apart from Giffen goods.

These results appear coherent with the preliminary findings provided by the analysis of consumption habits (see paragraph 3.1), which show that the respondents who frequently purchase organic EVOO are about ¾ of those consuming PDO/PGI EVOO (63% and 83%). This is the same ratio of belonging to classes 1 and 2 of the LCM analysis. Moreover, these results appear coherent with the preliminary findings of the analysis of consumption habits, showing that the origin of olives and/or olive oil appears a more relevant attribute of EVOO than organic certification, given the attention consumers pay to it. Finally, the analysis reported in paragraph 3.1 also highlights consumers’ limited consideration of the price of the EVOO bottle.

## 4. Discussion

This study aimed to investigate olive oil consumption behaviors in northeastern Italy, in particular with respect to five attributes: the country of origin (Italy, EU, or other countries), the presence (or absence) of PDO and PGI certifications, organic certification, leading brands, and price. We quantified the WTP for these attributes. Specifically, we attempted to measure the influence of various factors, such as organic certification and country of origin, on consumer purchase behavior, and to assess preference heterogeneity due to both observed and unobserved effects, as the unobserved effects could be relevant for olive oil [61].

Our findings point out a preference heterogeneity in the information perceived by olive oil consumers, identifying a number of unobserved sources of heterogeneity in their decision process. The presence of preference heterogeneity among the participants helped us to better explain underlying mechanisms driving individual choice.

This research reveals a strong and positive preference for locally produced olive oil as mainly suggested in the literature [57,64,66,67,72,73,74,77]; in particular, Finardi et al. [73], Casini et al. [7], and Panico et al. [50] report that Italian origin has a large positive effect on Italian EVOO buyers. Perhaps due to perceived negative or potentially negative effects on health of a number of accidents caused by contaminated food, the respondents related their preference for local products to their greater perceived safety when compared with foreign ones. This confirms the findings by Del Giudice et al. [45] on the strategic role played by knowledge, on the part of the consumer, of the oil’s origin. However, this result is not obvious, as, for example, Mtimet et al. [51] demonstrate that in Tunisia, the region of origin attribute had no significant effect on respondents’ purchasing decisions. In addition, Mtimet et al. [65]) state that Japanese consumers preferred olive oil of Mediterranean or Tunisian, rather than Italian, origin.

Yangui et al. [97] report that respondents did not grant superior value to EVOO organic attribute, perhaps as a consequence of the belief that olive oil is a healthy and natural product, regardless of its organic status. Similarly, our findings suggest that respondents may benefit from deeper information about organic methods of production. On the contrary, while Erraach et al. [72] demonstrate that price and PDO certification were the attributes that most affected consumer preferences, these appeared to be less relevant in our study. These attributes, which could however be appreciated by specific population segments, are not the only characteristics the respondents looked for.

The results of the LCM segmentation suggest the presence of a consumer segment who is positively impacted by the price coefficient. This is not a novelty: in fact, according to Romo-Muñoz et al. [98], respondents often consider price as a realistic and reliable quality clue.

Our study reveals useful information, which could potentially come in handy for different stakeholders. The results generally confirm expectations built on existing literature and may support the adoption of more efficient and complete marketing strategies by EVOO producers and distributors.

Indeed, a better knowledge of what olive oil consumers need and deem important and valuable is essential to both communicate salient features of existing lines of products and properly direct the selection and development of new lines according to customers’ needs. At the same time, stakeholders involved in the EVOO industry can identify prejudices and misconceptions on the products and subsequently intervene and educate consumers. Better-informed customers would take more informed and rational decisions with mutual gains for them, in terms of satisfaction, and the industry as a whole, which would be pushed towards efficiency and qualitative improvement. In general, it is necessary to further reduce the information asymmetries that hinder market efficiency [82,83]. In particular, it appears important to inform consumers more about the characteristics of the products and the meaning of the certifications and to disseminate more nutritional recommendations according to international and national guidelines.

Finally, it should be noted that the sustainability of the olive oil supply chain is a key element in the context of the growing worldwide attention to the healthiness of the Mediterranean diet. Therefore, the olive oil systems can play an important role within the Mediterranean diet as “a driver of sustainable food systems within the strategies of regional development and on that of traditional local products, since quantitative food security must also be complemented by qualitative approaches” [32] (p. 40).

In this respect, the development of a sustainable food system is accompanied by local sustainable development policies that take into account different aspects of sustainability, not least the cultural heritage of rural world and the agricultural landscape [33,99] according to an endogenous development model. [100]. In relation to this last aspect, it should be pointed out that the sustainability of the local food system at the base of the Mediterranean diet must be related to the production area. Otherwise, in a global context characterized by growing international trade, the environmental impact aspects should be assessed by including transport, logistics, and distribution activities according to a “Farm to Fork” approach [11]. This perspective would require a different analysis approach for a different research scenario and highlight potential limitation of this survey focused on domestic consumption.

In accordance with the sustainable food system linked to the MD, organic certification is only one of the attributes that can be exploited together with other environmental and socioeconomic characteristics, for instance, the characteristic of a typically Italian and local product with the certification of origin of the raw material (100% Italian olive oil) and compliance with GI certification (PDO, PGI), which are particularly appreciated and demanded by consumers, according to the results of this study.

Therefore, the sustainability of an olive oil system should be analyzed by taking into account not only one dimension of sustainability, but its overall multidimensional attributes within the space of a local development model, and integrating the endogenous local development model with a healthy and sustainable diet model.

Nevertheless, this survey has a number of limitations, which suggest future research developments. First of all, the sample of respondents was characterized by a geographically limited area (mainly from northeastern Italy) and a sociocultural profile that it is not representative of the entire Italian population. Second, data collection took place in a very traditional way (face-to-face interview); hence, the adoption of other data collection methods can influence the findings.

Therefore, notwithstanding the relevance and usefulness of our findings, the need to refine results calls for further development of research and advance of knowledge on this topic. Further research should also take into account the representativeness of the sample and consider alternative data collection methods. Moreover, even though the attributes and levels used in this study were carefully selected, findings may have differed with the inclusion of other characteristics, such as carbon, water, or ecological footprint certifications, eco-packaging, and vegan certification, whose demand is growing [101]. Finally, being that Italians are traditional EVOO consumers often fond of specific products or labels, the extension of the results to less mature markets may be difficult, if not misleading. Finally, our findings might not be directly extended to foreign EVOO markets: in fact, in spite of the common ground of the IFOAM standards, organic farming regulations vary across nations, together with consumers’ familiarity, understanding, and trust; therefore, further replications of our study in other contexts are highly desirable to estimate variations in consumers’ preferences for and attitudes towards organic EVOO.

## Figures and Tables

**Figure 1 foods-10-00994-f001:**
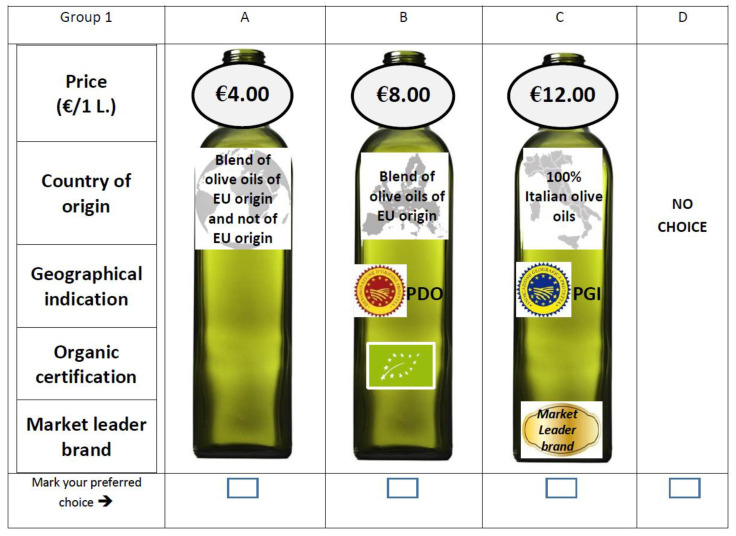
A choice set example for one of the six sets proposed.

**Table 1 foods-10-00994-t001:** Attributes and levels for olive oil in Italy.

Attribute	Description	Level
Price	The three levels of the price per bottle (1 L).	€4.00, €8.00, €12.00
Country of origin (COO)	The country where olives were produced. It appears on the label.	100% Italian olive oils, blend of olive oils of EU origin, blend of olive oils of EU origin and not of EU origin
Geographical indication (GI)	Label that indicates whether the product has a GI certification.	PDO, PGI, None
Organic	Organic certification label.	Yes, no
Market leader brand	The presence of a market-leading top brand, if any.	Yes, no

**Table 2 foods-10-00994-t002:** Statistical indicators for model comparison.

	LCM-2	LCM-3	LCM-4	LCM-5
LL	−5866.859	−5585.728	−5579.741	−5620.626
AIC	1.915	1.827	1.828	1.844
BIC	1.934	1.855	1.866	1.892
HQIC	1.922	1.837	1.841	1.861
McFadden pseudo *R*^2^	0.311	0.344	0.345	0.340

**Table 3 foods-10-00994-t003:** MNL and LCM results.

	MNL	LCM					
		Class 1		Class 2		Class 3	
Variable	Coeff. (S.E.)	Coeff. (S.E.)	WTP (€/l)	Coeff. (S.E.)	WTP (€/l)	Coeff. (S.E.)	WTP (€/l)
ASC	1.28	4.01	/	−1.52 (0.15) ***	/	5.97 (0.26) ***	/
Price	(0.08) ***	(0.59) ***	/	−0.07 (0.01) ***	/	0.24 (0.02) ***	/
COO: Italy	−0.01	−0.10	/	0.94 (0.09) ***	13.35	2.70 (0.21) ***	/
COO: EU	(0.01) **	(0.10)	/	0.83 (0.06) ***	11.80	0.52 (0.23) **	/
Organic	1.82	5.70	/	0.10 (0.11)	/	−0.18 (0.24)	/
Market leader brand	(0.06) ***	(0.69) ***	/	0.19 (0.09) **	2.67	−0.52 (0.21) **	/
GI: PGI	0.83	1.10	/	0.47 (0.08) ***	6.69	0.85 (0.24) ***	/
GI: PDO	(0.06) ***	(0.57) **	/	−0.26 (0.03) ***	−3.67	−0.44 (0.08) ***	/
Average probability		1.59		0.46		0.24	

*** Significant at a 95% conf. level; ** significant at a 90% conf. level.

## Data Availability

The data that support the findings of this study are available from the corresponding authors upon reasonable request.

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
