# Peer review of "Consumer Preferences for Origin and Organic Attributes of Extra Virgin Olive Oil: A Choice Experiment in the Italian Market"

_foods, 2021, doi:10.3390/foods10050994_

Round 1

Reviewer 1 Report

The paper topic and it’s surveyed data are of global interest, however, the key terms are not well connected and justified. 
In terms of the Introduction, I’d suggest further literature review on perceived quality of organic foods and values of locally produced food. The distinctions of olive oil and EVOO should be stated in terms of production, grading standards, nutritive values in order to make links to MD and sustainability. Organic certification schemes (e.g. Soil Association-UK, NOP-USDA) set out slightly different criteria which could also affect consumer’s perceived quality of organic produce.

In terms of the Methodology, qualitative information from focus group discussion (mentioned in this part) should be included to complement and explain the questionnaire design. The questionnaire structure, where branched questions were applied, answer choices and scales used are of importance as well as sampling plan and how the respondents were recruited. The detailed information would make a clear connection to statistical analyses of the results - considering the classification of EVOO buyers, identification of key factors and discussion of results are mainly based on this surveyed data.

Reviewer 2 Report

 The statistical analysis is sound and justified and also the methodology is adequately described. As a proposal, an application of Cluster Analysis simultaneously with the used methodology, maybe could confirm the results with a sounder way. One drawback is that the results of the study have been not compared with other similar published studies. Lastly, I consider that the study provides sufficient background and includes all relevant references

  • Specific comments

Concerning the specific comments:

Line 85. I recommend the word “reputation” to be replaced by the word “perception”. It is more clear term. Alternatively, the first term must be specified with a more precise way.

Lines 201-203. There is not information about the sample selection and sampling method. Give details about the afore mentioned issues.

Lines 268-395. In this section results of similar published studies could be added. By this way, the findings of the study could be reinforced.

Line 418. The comma (,) between the words ……“as, for example”…. is unnecessary.

Line 421. The comma (,) between the words ……“Italian, origin”…. is unnecessary.
